# Picosecond time-resolved photon antibunching measures nanoscale exciton motion and the true number of chromophores

Gordon J. Hedley [1,5✉], Tim Schröder [2,5], Florian Steiner [2], Theresa Eder[3], Felix J. Hofmann[3], Sebastian Bange[3], Dirk Laux[4], Sigurd Höger[4], Philip Tinnefeld [2], John M. Lupton [3] & Jan Vogelsang [3✉]

The particle-like nature of light becomes evident in the photon statistics of fluorescence from single quantum systems as photon antibunching. In multichromophoric systems, exciton diffusion and subsequent annihilation occurs. These processes also yield photon antibunching but cannot be interpreted reliably. Here we develop picosecond time-resolved antibunching to identify and decode such processes. We use this method to measure the true number of chromophores on well-defined multichromophoric DNA-origami structures, and precisely determine the distance-dependent rates of annihilation between excitons. Further, this allows us to measure exciton diffusion in mesoscopic H- and J-type conjugated-polymer aggregates. We distinguish between one-dimensional intra-chain and three-dimensional inter-chain exciton diffusion at different times after excitation and determine the disorder-dependent diffusion lengths. Our method provides a powerful lens through which excitons can be studied at the single-particle level, enabling the rational design of improved excitonic probes such as ultra-bright fluorescent nanoparticles and materials for optoelectronic devices.

[1] School of Chemistry, University of Glasgow, Glasgow, UK. [2] Department Chemie and Center for NanoScience (CeNS), Ludwig-Maximilians-Universität München, München, Germany. [3] Institut für Experimentelle und Angewandte Physik and Regensburg Center for Ultrafast Nanoscopy (RUN), Universität Regensburg, Regensburg, Germany. [4] Kekulé-Institut für Organische Chemie und Biochemie, Universität Bonn, Bonn, Germany. [5] These authors contributed equally: Gordon J. Hedley, Tim Schröder. ✉email: Gordon.Hedley@glasgow.ac.uk; Jan.Vogelsang@physik.uni-regensburg.de

In a wide range of fluorescent nanoparticles such as conjugated polymers, semiconductor quantum dots, perovskite nanoparticles, light-harvesting complexes and many other natural or synthetic multichromophoric nanoparticles (mcNP), multiple excitons can exist simultaneously and in close proximity to each other[1–10]. The number of chromophores as well as their interactions through exciton diffusion and annihilation processes are key parameters to describe the photophysical characteristics of mcNPs such as brightness[11,12], photoluminescence (PL) lifetime, exciton harvesting efficiency[13] and photostability[12,14], all of which are also important for the performance of materials in optoelectronic devices. Photon antibunching has been used to count chromophores[15–17]; however, this is typically not viable when exciton diffusion and singlet-singlet annihilation (SSA) occur as illustrated in Fig. 1a. Single-photon emission from mcNPs has been interpreted as evidence of long-range inter-chromophore interactions in a number of large multi-chromophoric systems[1,2,8,18–22]. However, in these cases information about the number of physical chromophores in the mcNPs is lost. Here, we demonstrate that picosecond time-resolved antibunching (psTRAB) can be used to disentangle information on the number of physical chromophores and exciton diffusion and annihilation processes. psTRAB exploits the fact that exciton diffusion and annihilation are time-dependent processes. Fingerprints of these processes are thus concealed in the PL photon stream of antibunching experiments under pulsed excitation[6,23].

The degree of single photon emission is commonly measured by two photodetectors in a Hanbury Brown and Twiss (HBT) geometric configuration and is therefore sensitive to two-photon events. With this technique, it is either possible to count the number of chromophores, provided that SSA is neglected, or to measure the SSA rate if the exact number of chromophores is known. In practical situations, neither the number of chromophores nor the SSA rate are usually known for mcNPs, which severely limits the usefulness of this conventional technique.

With psTRAB, we analyse the photon stream of antibunching experiments with pulsed excitation by grouping photons with respect to their arrival time after the laser pulse and cross-correlating them to determine the probability of consecutive emission of two photons. Immediately after a laser pulse, SSA has not yet occurred and the emitted photons exhibit photon statistics corresponding to the number of physical chromophores present. As exciton diffusion and annihilation begin to dominate, the number of independent emitters decreases. Thus, the time-dependence of the photon statistics synchronised by the laser pulse reports on (i) the number of physical emitters present and (ii) the time evolution of exciton diffusion and annihilation.

To demonstrate the psTRAB technique, we have used DNA origami to construct mcNPs with a known number of chromophores and well-defined spacing between them to accurately measure annihilation and benchmark our method. We then measure psTRAB of mesoscopic deterministic aggregates of conjugated polymers—the building blocks of films used in optoelectronic devices[2]. There we find that during the first 250 ps after excitation, diffusion of excitons mainly occurs between one and two dimensions, both along the polymer backbone and between π-stacked chains. The diffusion then becomes three-dimensional at later times, with an order-of-magnitude difference in the rate of annihilation between ordered H-type aggregates and disordered J-type aggregates. We can also extract the exciton diffusion lengths using the unique knowledge psTRAB gives on the number of independent chromophores present.

Our approach exploits the ability of modern time-correlated single-photon counting (TCSPC) hardware to record the absolute arrival time of a photon on each detector, both with respect to the

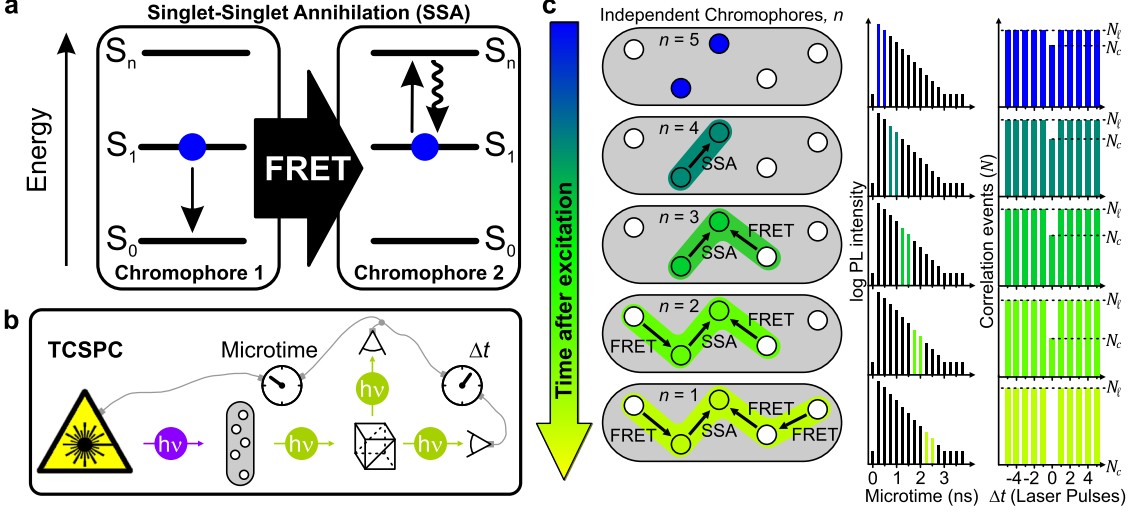

**Fig. 1 Picosecond time-resolved antibunching (psTRAB). a** Singlet–singlet annihilation (SSA) of a singlet exciton, $S_1$, on chromophore 1 by Förster resonance energy transfer (FRET) to an exciton on chromophore 2, which excites it into a higher excited state, $S_n$. Subsequently, chromophore 2 relaxes by internal conversion into its first excited state, $S_1$. Thus, the exciton (blue dot) on chromophore 1 is annihilated. **b** Principle of time-correlated single-photon counting (TCSPC) combined with a Hanbury Brown and Twiss photon correlator. A pulsed laser (purple) excites a multichromophoric nanoparticle (mcNP) (grey area). The statistics of the PL photon stream (green) are analysed by cross-correlating the signal of two photon detectors. TCSPC yields the time difference between excitation and emission events, i.e. the "microtime", and the time difference between consecutive emitted photons, $\Delta t$, as determined by the repetition period of the pulsed laser. **c** On the left, five chromophores (discs) in an mcNP are shown schematically with two singlet excitons (coloured discs), which after excitation can diffuse by site-to-site hopping, i.e. by homo-FRET and annihilate by SSA as a function of the excited-state lifetime. The overall PL decay, constructed from the microtimes, is shown in the centre, with the corresponding binned arrival time windows of photons used to construct the antibunching histograms stated in the right-hand column. The ratio, $N_c/N_\ell$, of the number of correlation events in the central peak at $\Delta t = 0$, $N_c$, versus those in the lateral time-lagged peaks, $N_\ell$, allows us to determine the number of independent chromophores, $n$. As excitons diffuse through homo-FRET and annihilate through SSA, $n$ drops with time.

start of the experiment, but also with respect to the last laser pulse (denoted as the microtime) as shown in Fig. 1b. As an example, consider a nanoparticle with five physical chromophores as depicted by the white discs in Fig. 1c. Absorption of a short pulse of light will create a Frenkel exciton (blue disc). The exciton can hop from one chromophore to another, e.g. by homo-FRET[24,25], in a process referred to as exciton diffusion[26]. Now, if we consider the case where two excitons are created by the same excitation pulse, this hopping allows the excitons to move so that they become adjacent to each other and can annihilate by SSA[19,20]. This process has a strong distance dependence due to the underlying FRET mechanism by which SSA occurs and is often hard to study in a quantitative manner[27]. By inspecting individual mcNPs on a confocal microscope with two single-photon detectors (Fig. 1b) combined with TCSPC we measure the correlation events, $N$, dependent on the difference in photon arrival times, $\Delta t$, between photon events. We are thereby sensitive to the presence of two excitons in the mcNP. A histogram of $\Delta t$ delay times in integer units of the excitation-pulse period $T$ shows the number of photon detection coincidences from either one excitation pulse or from two separate excitation pulses (Fig. 1c, right column). The ratio of the magnitude of the central peak at $\Delta t = 0$ to that of the lateral peaks, $N_c/N_\ell$, provides a measure for the number of independent chromophores, $n$, provided that the background can be accounted for (see Supplementary Eq. 1 for details on the background correction) according to[16]

$$n = 1 \Big/ \left( 1 - \frac{N_c}{N_\ell} \right) \qquad (1)$$

By analysing the statistics of the PL photons detected at different time intervals after photoexcitation (panel c, second column), we can construct corresponding picosecond-resolved histograms of the photon statistics and thus measure how many independently emitting chromophores exist on a particular timescale. This is illustrated schematically in Fig. 1c for a 5-chromophore mcNP. The left column depicts the evolution of randomized typical examples of such independent chromophores after a single laser excitation event, whereas the histograms in the middle and right columns are an accumulation of multiple excitation cycles to show the time-averaged result. At early times after excitation (panel c, first row), the two excitons contributing to $N_c$ events (blue discs) have had no time to interact or move via homo-FRET to neighbouring chromophores. From the photon coincidence histogram (right panel) we obtain a value of $n = 5$ with Eq. (1). At a later time (panel c, second row), an exciton on a neighbouring physical chromophore may have, for example, interacted through SSA, and consequently excitation of such chromophores thus does not contribute to $N_c$ anymore, and we obtain $n = 4$ independent chromophores accordingly. These diffusion/SSA processes continue as a function of time, reducing the number of independent chromophores that could support the second exciton. Ultimately, at late times after the excitation pulse, only single photons can be detected because excitons on any other physical chromophore would have had enough time to diffuse and annihilate, yielding $N_c = 0$ and $n = 1$ (panel c, last row). This evolution of the photon statistics and the corresponding number of independent chromophores with time gives us a metric for the effective rate of exciton decay and provides direct microscopic insight into exciton annihilation and diffusion in mcNPs.

## Results

**Exciton annihilation in DNA origami nanoparticles.** To explore the fundamental nature of exciton diffusion and SSA it is desirable to have the best possible control over the number of dye molecules and their spatial position in the mcNP. The dyes need

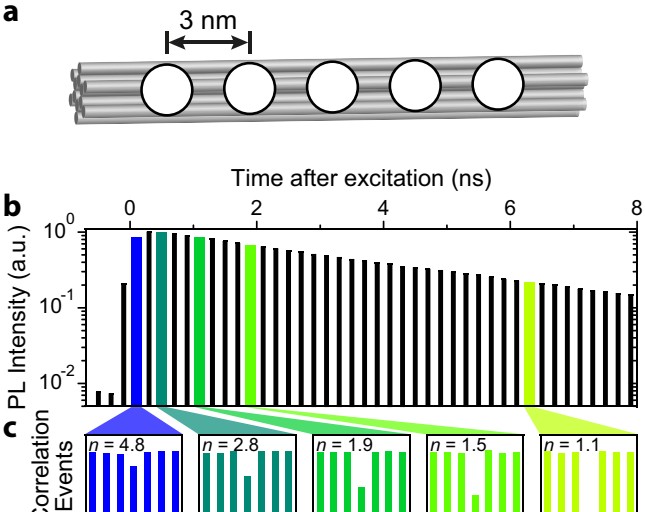

**Fig. 2 Tracking exciton diffusion and annihilation in space and time on a well-defined multichromophoric DNA origami structure. a** Schematic of a short part of a 225 nm long 12-helix-bundle DNA origami structure with 6 inner and 6 outer helices. Five dyes (white discs) can be positioned at 3 nm spacing from each other. **b** Measured photoluminescence (PL) decay of a single DNA origami structure with five ATTO647N dye molecules. A single-exponential decay is observed with a PL lifetime of ~4.2 ns. Panel **c** shows five corresponding photon statistics histograms for different microtime bins (0–200, 200–400, 800–1000, 1600–1800 and 6400–6600 ps) in terms of the photon correlation events, $N$, of the two photodetectors. Data were accumulated for 54 single mcNPs. The number of independent chromophores, $n$, determined from the correlation histogram for each microtime bin is stated above the histograms.

to be within distances to each other corresponding to the range of FRET of ~1–10 nm. We have therefore turned to the method of three-dimensional DNA origami to construct highly defined mcNPs. Similar structures have been used previously to study motor proteins and to characterize super-resolution microscopy techniques[28,29], and are modified here for our needs. The sketch in Fig. 2a shows a short section of a 12-helix bundle with 6 inner and 6 outer helices. The total length of this DNA origami structure is ~225 nm (transmission electron microscopy (TEM) images and structure are shown in Supplementary Figs. 1 and 2). Five labelling positions separated by ~3 nm each are available in the centre of this modular structure.

Based on this 12-helix bundle DNA origami structure, we designed seven different structures with different numbers of dyes and different distances between the dyes (see Supplementary Information for details of DNA origami structures). For the dye we chose ATTO647N, which is highly photostable and bright in the presence of a reducing and oxidizing system (ROXS)[30]. The origami structures were examined on a custom-made confocal fluorescence microscope as described in the "Methods" section (a typical PL transient is shown in Supplementary Fig. 3)[12]. We begin discussing the mcNP with all five dye attachment positions filled with a dye. Figure 2b displays a histogram of photon arrival times, i.e. microtimes, in steps of 200 ps following pulsed excitation with a 636 nm laser. We note that the step size also defines the timing error on the $x$-axis of the plot. This value of 200 ps was chosen according to the budget of photons available to construct the histograms of photon statistics in Fig. 2c. It is necessary to make a trade-off between the timing resolution of the $x$-axis and the noise in the photon statistics histograms. This

trade-off depends on the experimental circumstances, i.e. the photon budget which is available. The PL decay is single exponential with a lifetime of 4.2 ns, which is typical for this dye attached to DNA and implies that no strong interchromophoric interactions occur[12].

For this five-dye sample we select 200 ps time windows from the microtime histogram (coloured bars) and calculate the photon statistics for each bin as shown in Fig. 2c. We used the peak of the instrument response function (see Supplementary Fig. 4) to determine zero microtime in the calculations. According to Eq. (1), we estimate the number of independent chromophores, $n$, in the first 200 ps after excitation to be ~4.8, very close to the expected starting value of 5. Between 200 and 400 ps, $n$ drops to ~2.8 and reaches ~1.1 between 6400 and 6600 ps. The photons emitted by the five-chromophore structure at the latest times show almost complete antibunching. In total, photon events of 54 individual mcNPs were accumulated to obtain enough correlation events for this analysis. Photobleaching and blinking of individual dye molecules during the measurement period will impact the overall photon statistics. For this reason, only the first 5 s of each measurement were evaluated, and only if the overall PL intensity was constant to within 10% over this time. Additionally, while photobleaching and blinking has an influence on the overall strength of photon antibunching, it has no impact on the decay of $n$ with microtime. For example, we indeed obtain the expected starting value of 5 for early microtimes, implying that the measurement is not affected by photobleaching and blinking. The five histograms in Fig. 2c reveal the timescale on which the excitons annihilate with each other to lower the number of independently emitting chromophores from five to one. We note that the fact that the number of chromophores inferred at the earliest times is slightly lower than the expected value of five can be explained by SSA having already occurred during the first 200 ps. One immediate conclusion of this method is that the number of dyes can be measured in an mcNP directly, even if the dyes are not emitting independently. Such knowledge is crucial in quantitative spectroscopic methods[17,31]. A further crucial observation is that, in contrast to ensemble measurements[32], the PL decay retains its monomolecular single-exponential form even though SSA clearly occurs. This is a particularly important observation because the non-exponentiality of ensemble PL decays, i.e., a bimolecular decay, is generally used to extract exciton encounter rates to infer diffusion lengths. In the ensemble, this approach only works at very high excitation fluences which are far from the population densities relevant to devices. However, it is crucial to realize that SSA always occurs, even at the lowest excitation fluences, because exciton diffusion always occurs. Our photon correlation technique is sensitive precisely and only to these rare events of double-chromophore excitation, which can be reached at very low fluences at the cost of extended integration times. The detection of these rare events is ultimately limited by the background photons, e.g. the dark count rate of the photo detectors.

Having established that we can recover the number of dyes in an mcNP with our method, we now apply this approach to different DNA origami structures to examine the dynamics of the SSA mechanism in detail. Figure 3 plots the number of independently emitting chromophores $n$ for each 200 ps time gate versus the corresponding microtime for seven different DNA origami structures. We start with the simplest model system with only one dye (dark grey dots in panel a). Except for the first two data points, these values are constant at $n = 1.02$, which is expected for a single dye. This value is close to unity and only limited by the signal-to-background ratio (SBR) as discussed in Supplementary Fig. 5 and ref. [15]. The fast decay in the first two data points originates from multiple excitations of the dye within

the same laser pulse of ~80 ps width[33]. Now we introduce a second dye at a distance of ~12 nm (panel b, light grey dots), which should be far enough away to prevent SSA between the excitons. Indeed, the data can be described with a constant $n$ of $1.85 \pm 0.01$, which is slightly below the expected value of two, most likely because of slightly different PL intensities of the two dye molecules at the different binding sites of the DNA origami structure. Crucially, again, no decay of $n$ is observed for this sample, implying a negligible exciton annihilation rate.

Next, we examine the more interesting cases, where we build structures with two dyes sufficiently close to each other such that SSA can occur. The red and orange dots in Fig. 3b display the data measured on structures carrying two dyes at ~3 and ~6 nm spacing. $n$ starts out slightly below the expected value of two for both samples, and a decay during the first 2 ns down to $n = 1.02$ is observed for the 3 nm sample. These datasets are accurately described by a single-exponential model of the number of independently emitting chromophores,

$$n(t) = \{y_0 - [A \cdot \exp(-k_{SSA}\, t)]\}^{-1} \qquad (2)$$

with the offset, $y_0$, amplitude, $A$, and the exciton annihilation rate, $k_{SSA}$ (see "Methods" for a derivation of Eq. 2). The overall number of physical dyes present in the structure is then given by $n_{dyes} = (y_0 - A)^{-1}$. In Fig. 3b, we extract $k_{SSA} = 1.72 \pm 0.06$ ns$^{-1}$ for the two dyes separated by 3 nm and $k_{SSA} = 0.06 \pm 0.01$ ns$^{-1}$ for the dyes separated by 6 nm, with $n_{dyes} = 1.8 \pm 0.03$ in both cases. As expected, $k_{SSA}$ drops significantly when doubling the distance between the two dyes, indicating that we are in the important regime where SSA is controlled by FRET and therefore by dye spacing. Subsequently, we placed three dyes separated by ~6 nm each (Fig. 3c, cyan dots). Fitting with Eq. 2 yields $k_{SSA} = 0.06 \pm 0.01$ ns$^{-1}$ and $n_{dyes} = 2.7 \pm 0.1$, which is consistent because we expect no SSA between the left-most and right-most dyes, and the same SSA rate for the neighbouring dyes as in panel b.

Upon moving the three dyes closer to each other, now only separated by 3 nm (Fig. 3c, blue dots), Eq. (2) is no longer sufficient to describe the time evolution of $n$ since next-nearest neighbour interactions arise. We therefore used an analogous biexponential model of SSA, with a fast rate for neighbouring dyes and a slow rate, which combines direct annihilation of next-nearest-neighbouring dyes and exciton hopping with subsequent annihilation of neighbouring dyes, to describe the blue dataset in panel c,

$$n(t) = \{y_0 - [A_1 \exp(-k_{SSA,1} \cdot t) + A_2 \exp(-k_{SSA,2} \cdot t)]\}^{-1} \quad (3)$$

We derive from this dynamics an average amplitude-weighted SSA-rate $\langle k_{SSA} \rangle = (A_1 k_{SSA,1} + A_2 k_{SSA,2})/(A_1 + A_2) = 0.98 \pm 0.09$ ns$^{-1}$ (see Supplementary Information for complete fitting results in Supplementary Table 2) and a number of dyes, $n_{dyes} = (y_0 - (A_1 + A_2))^{-1} = 2.9 \pm 0.1$. Finally, for the DNA origami structure bearing all five dyes (Fig. 3d, violet dots), we extract $\langle k_{SSA} \rangle = 0.72 \pm 0.07$ ns$^{-1}$ and $n_{dyes} = 4.7 \pm 0.2$ by using Eq. (3).

The crucial observation is that at long microtimes, $n$ decays to 1 for all samples with $k_{SSA} > 0$. This is particularly intriguing for the five-dye sample, where we would anticipate the case in which two excitons remain on the left-most and right-most dyes. According to the experiment with two dyes placed 12 nm apart (panel a, light grey dots), no direct SSA should occur in this case. However, the fact that the five-dye sample still decreases down to only one emitting independent chromophore, rather than two, allows us to conclude that exciton hopping, i.e. exciton diffusion, occurs between the five dyes. We note that all measurements of the DNA origami samples were conducted in buffered solution

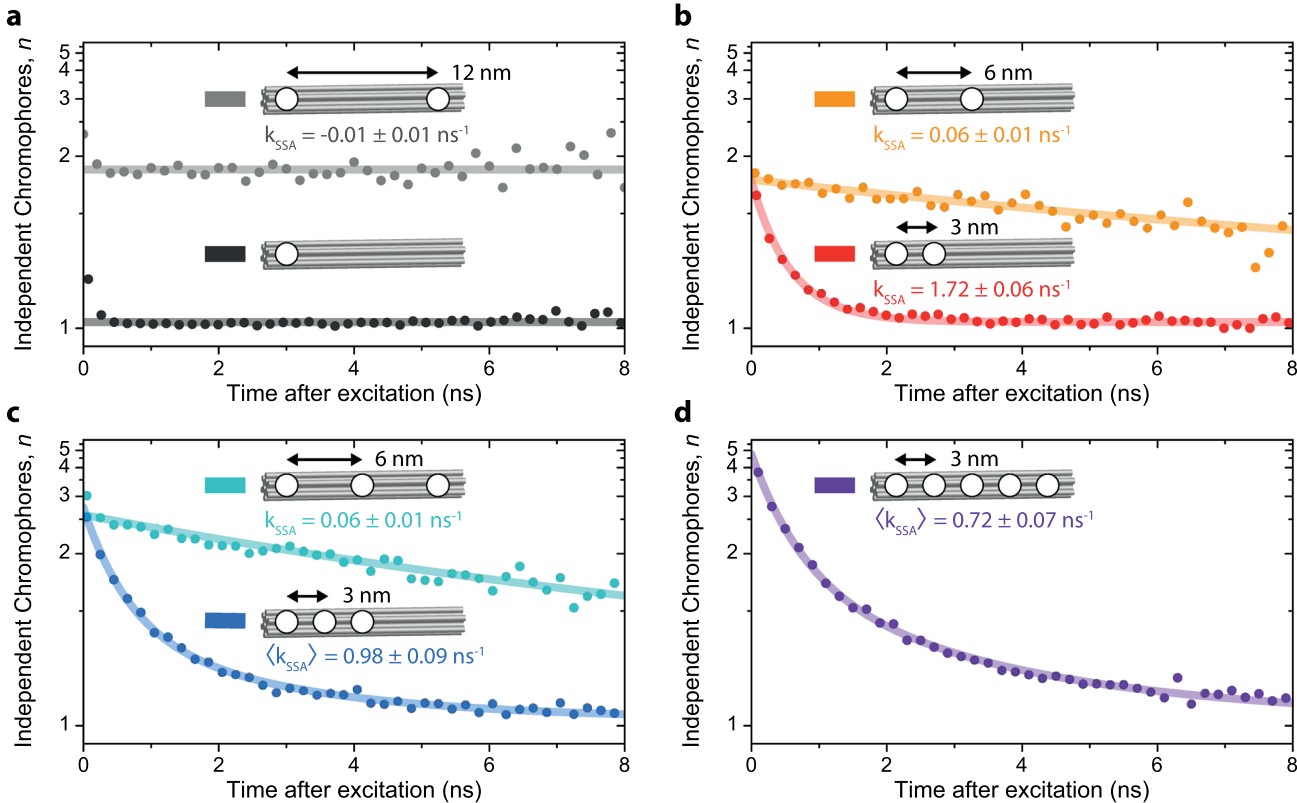

**Fig. 3 Exciton annihilation in multiple well-defined mcNPs.** Evolution of the number of independently emitting chromophores, $n$, as a function of the time after excitation for seven different structures. The structures have different numbers of dyes attached with different spacings between them. The $y$-axes are reciprocal for better comparison between the graphs. **a** One dye (dark grey) and two dyes separated by 12 nm (light grey). **b** Two dyes separated by 3 nm (red) and 6 nm (orange). The curves superimposed are described by a monoexponential model of exciton annihilation (Eq. (2)). **c** Three dyes separated by 6 nm (cyan) and 3 nm (blue). The cyan curve is described by Eq. (2), but the blue points follow a biexponential decay with an average rate $k_{SSA}$ (Eq. (3)). **d** Five dyes separated by 3 nm. The curve is described by the biexponential decay of Eq. (3). Between 54 and 98 single mcNPs were measured individually, and the photon statistics of each measurement accumulated to obtain each curve. All measurements were performed under oxygen removal and with a reducing and oxidizing system (ROXS) present to ensure photostabilization[30]. Each particle was measured for only 5 s so that photobleaching and spectral shifts were negligible.

and consequently, the dyes were free to rotate on the DNA origami. We therefore neglect the possibility of a particular preferred orientation of the transition-dipole moments arising. However, this approximation is no longer valid for mcNPs, which are fixed in space, e.g., set inside a solid matrix. Here, the transition-dipole moment orientation can have a significant impact on the SSA rate, i.e., the morphology plays a crucial role on the dynamics of psTRAB. This conclusion offers a motivation to study morphologically different mcNPs in which significant exciton diffusion arises.

**Exciton diffusion in conjugated polymer aggregates**. To examine exciton diffusion in conjugated polymers in the meso-scopic size regime, aggregates of chains were grown with distinct electronic and structural properties. These structures are formed by two poly(para-phenylene-ethynylene-butadiynylene) (PPEB)-based conjugated polymers (Fig. 4a). With a small variation of the alkyl side-chains, ordered aggregates with either H-type inter-chromophoric coupling (PPEB-1, lilac) or disordered aggregates with J-type intrachromophoric coupling (PPEB-2, brown) can be grown by solvent vapour annealing[18]. Samples were prepared as described in ref.[18], yielding individual small aggregates isolated in poly(methylmethacrylate) (PMMA) and measured on a confocal fluorescence microscope as reviewed briefly in the "Methods" section and described elsewhere[34]. 631 single aggregates of PPEB-

1, each comprising on average approximately 54 chains, and 705 aggregates of PPEB-2 (each ~9 chains, see Supplementary Figs. 6 and 7 and discussion thereof in the Supplementary Information), were grown and measured individually. Only the first 5 s of each PL trace were evaluated (see Supplementary Fig. 8 for examples of PL traces of the H- and J-type aggregates), provided that the PL intensity was constant to within 10%. Following the above procedure, $n(t)$ was determined using psTRAB as shown in Fig. 4b (Supplementary Fig. 10 shows the corresponding photon anti-bunching histograms). We use different widths of time-windows to generate the evolution of $n(t)$, with 3 ps chosen at early times, increasing to 80 ps (in the H-type aggregates) and 160 ps (in the J-type aggregates) at later times. We observe a clear decay of $n$ with time, signifying excited-state interactions primarily due to SSA. We note that this measurement is independent of the excitation intensity in this region of excitation densities as discussed in Supplementary Fig. 9. A substantial difference between the decay dynamics exists for the two aggregates. For the H-type aggregates, $n$ drops rapidly over the first 250 ps and then continues before levelling off at ~2000 ps. The J-type aggregates show a smaller initial fast drop, followed by a slower linear decay before levelling off at a slightly higher value of $n$ at times >2000 ps.

First, we note that, in analogy to the DNA origami model system in Fig. 3d, the decay of $n$ with time constitutes a signature of exciton annihilation mediated by exciton diffusion. Because diffusion is now likely to dominate, however, the dynamics

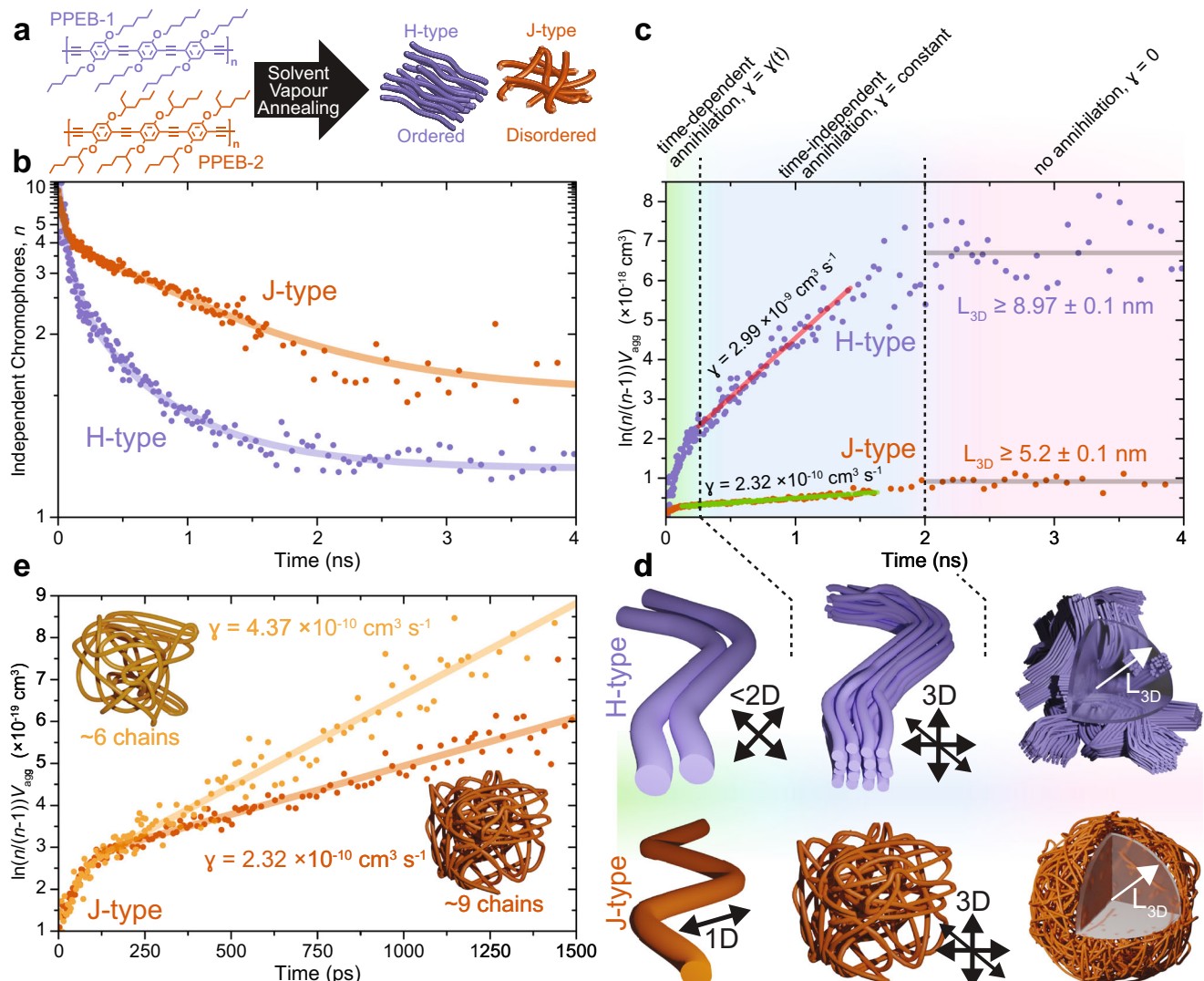

**Fig. 4 Following exciton diffusion dynamics in isolated conjugated polymer aggregates. a** PPEB polymers that are grown into H-type (PPEB-1, lilac) or J-type (PPEB-2, brown) aggregates by solvent vapour annealing. **b** Temporal evolution of the number of independent chromophores in the aggregate, determined by psTRAB, for the H-type and J-type aggregates. A significant difference in the dynamics is observed, with lines being guides to the eye. **c** Plotting of the time-dependent quantity $\ln\left(\frac{n}{n-1}\right) \cdot V_{agg}$ for the H- and J-type aggregates. The gradients of the curves correspond to the exciton annihilation rate, $\gamma$, of diffusion-controlled annihilation. Three regions of the dynamics are identified: at early time (0–250 ps), $\gamma$ is time-dependent; at intermediate times (250–2,000 ps) $\gamma$ is constant and tenfold higher in the H- compared to the J-type aggregate; and at late times (> 2000 ps) $\gamma$ is zero. These regions are interpreted in **d**, indicating that early-time diffusion is one- or less than two-dimensional and intermediate time diffusion is three-dimensional; at late times annihilation ceases because the exciton density is too low. The latter range provides a lower limit of the three-dimensional diffusion length, $L_{3D}$. In **e** the nine-chain J-type aggregate shown in panel **c** (brown) is compared with a smaller six-chain J-type aggregate (orange). The gradient ($\gamma$) is a factor of two smaller in the larger aggregate, indicating that three-dimensional diffusivity is reduced when more chains are present. This reduction is consistent with reduced ordering of the chains as the aggregate size grows, and thus reduced interchain coupling. The six-chain dataset is made up of 260 individually measured single aggregates.

generally cannot be fitted with one fixed $k_{SSA}$ rate. Instead, the annihilation is governed by a rate equation for a second-order reaction[35]. The clear difference between the H- and J-type aggregates indicates that the process of exciton diffusion is not the same in both of them. To examine this difference in a quantifiable manner, we plot the evolution with time of the quantity $\ln\left(\frac{n}{n-1}\right) \cdot V_{agg}$ as shown in Fig. 4c, where $V_{agg}$ is the calculated aggregate volume (see "Methods" for a full description of this equation and Supplementary Information for how the volumes were obtained). This allows us to quantify and compare exciton diffusion, as data plotted in this manner allows the instantaneous rate of bimolecular exciton annihilation, $\gamma$, to be determined from the slope and compared against ensemble equivalents. A linear

function signifies a constant, time-independent $\gamma$, whereas curvature implies that $\gamma$ has a time-dependence. Typically, in exciton annihilation measurements, the underlying excited-state decay has to be accounted for[36], complicating analysis in extracting diffusion relevant properties. The advantage of psTRAB is that we directly obtain a measure of the exciton diffusion and are thus uniquely sensitive to weak and slow diffusion. This contrasts with conventional ensemble measurements of the non-exponential decay in PL intensity, which require high exciton densities to see an appreciable effect of annihilation. It is important also to stress that psTRAB offers a unique way to observe the very rare circumstances where two excitons exist in a nanoscale object, and consequently to see how

the probability of them coexisting changes on the picosecond to nanosecond timescale as diffusion-assisted exciton annihilation occurs. The equivalent ensemble measurements of SSA require appreciable, i.e. measurable, fractions of excitons to annihilate with each other to be distinct from exciton luminescence where no annihilation has occurred. Consequently, as noted, psTRAB allows measurements of weaker and slower processes to be made than would otherwise be possible, with the measured photon coincidences in the PPEB H-aggregates typically ~300 parts per million, well below the overall luminescence signal's shot noise limit. With the data plotted as $\ln\left(\frac{n}{n-1}\right) \cdot V_{agg}$ as in Fig. 4c, for both H- and J-type aggregates three regions are identified. At early times (<250 ps) non-linear behaviour is observed, indicating that $\gamma$ is time-dependent. Exciton diffusion is therefore one- or less than two-dimensional[37]. At times 250–2000 ps, both aggregate types show linear behaviour, thus $\gamma$ is time-independent and the diffusion three-dimensional[38,39], with values of $\gamma$ found to be in the range of $10^{-9}$ to $10^{-10}$ cm$^3$ s$^{-1}$, in good agreement with typical conjugated polymers[32,39–41]. Finally, at times >2000 ps, $\gamma = 0$, i.e. annihilation has ceased as the exciton density is too low to support continued interactions.

The psTRAB results also allow insight into the nanoscale organization of material in the aggregates, as sketched in Fig. 4d. At early times, the time-dependent $\gamma$ indicates that exciton motion is one- or less than two-dimensional, most likely in the dispersive regime, and is therefore consistent with ensemble observations of annihilation on the timescale of a few picoseconds[42]. In the context of the H-type aggregate, this motion will be along the chains and across the interchain $\pi$-stack. This conclusion is in agreement with a high degree of chain alignment, evidenced by the PL intensity modulation depths determined when rotating the polarization of the exciting laser[18]. The J-type aggregate also shows time-dependent annihilation at early times. Here, however, simple one-dimensional motion will be favoured since strong intrachain coupling is dominant as evidenced by the J-type emission characteristics[18]. At later times, the time-independence of $\gamma$ indicates that exciton motion is three-dimensional in both aggregate types. $\gamma$ is an order of magnitude lower in this time region for the J- than for the H-type. This difference relates to the nature of chromophoric coupling and disorder in the aggregates. In H-type aggregates, chains with the smallest degree of disorder will show the strongest interchain electronic coupling, facilitating efficient three-dimensional diffusion. In J-type aggregates, in contrast, which do not show a high polarization anisotropy[18], chains are relatively disordered. Poor chain alignment will lead to weak interchain electronic coupling and a lower value of $\gamma$. Exciton diffusion is then limited by the random chain alignment that excitons encounter when diffusing. The impact of chain disorder on exciton diffusion can also be examined by comparing the psTRAB of the 9-chain J-type aggregate with a smaller one that comprises of ~6 chains shown in Fig. 4e. In the region where $\gamma$ is time-independent and three-dimensional diffusion dominates, $\gamma$ is almost a factor of two higher in the 6-chain aggregate, indicating increased order in the smaller aggregate which facilitates effective interchain site-to-site hopping. We also note that at early times (0–125 ps), in the J-type aggregates a significantly stronger time-dependent gradient of the psTRAB functionality is observed, consistent with fast one-dimensional exciton motion along the chain. We are cautious with regard to over-interpreting these data, however, since such exciton motion is likely to be much faster than the time resolution of our experiment. Indeed, we would expect the one-dimensional exciton motion along the chain in strongly coupled J-type aggregates to be higher than the two-dimensional diffusion along-chain and across $\pi$-stacks in H-aggregates, where intrachain coupling can be weaker[43,44].

Finally, at late times where $\gamma \rightarrow 0$, we enter the regime where the exciton density is too low to support continued annihilation. These conditions can be used to obtain a lower limit on the exciton diffusion length, $L_{3D}$. The rationale for this approach is simple: we know the volume of the aggregate and the number of independent chromophores that the aggregate can support when we can no longer measure annihilation occurring, i.e. when excitons no longer interact with each other. Division yields the volume that a single independent chromophore occupies, equivalent to the volume explored by an exciton. If diffusion is presumed to arise in a spherical volume in three dimensions, a diffusion length, $L_{3D}$, can be determined. The value will be a lower limit as the length is technically defined as the distance excitons diffuse in their lifetime rather than once the exciton density is too low to support continued interactions, but the difference between these two definitions will be small at these late times. We find lower limits of $L_{3D} \approx 9$ nm for the H-type aggregate and $L_{3D} \approx 5.2$ nm for the J-type aggregate, consistent with typical literature values for conjugated polymers[26,36,39,45]. The unique advantage of our chromophore-counting method is that the calculation of these values contains no presumptions other than the mass density of the aggregate. $L_{3D}$ is derived from simple observables and is only possible because we consider single objects at the discretised level of excitons and the resulting photon correlation.

## Discussion

Knowledge of the nanoscale organization of a material, the electronic coupling between chromophores, and energy transfer pathways is important in a wide variety of systems. In this work we have introduced a powerful method to quantify exciton–exciton annihilation and exciton diffusion in multi-chromophoric mesoscopic objects. This is achieved by resolving the fluorescence photon statistics on a picosecond timescale. Using deterministic DNA origami structures, we position dyes at specific distances from each other and obtain direct measurements of the rate of annihilation between two excitons and the true number of dyes. This accuracy is a direct consequence of utilizing two-detector coincidences that are sensitive to two-photon emission events. Our method can measure the annihilation rate $\gamma$ in well-defined structures and directly yields the number of physical dyes present in each sample. We stress that such chromophore counting is not possible with standard time-integrated photon-correlation measurements. The technique can be expanded to look at nanoparticles grown from multiple single conjugated-polymer chains. In these polymer aggregates, SSA is governed mainly by exciton diffusion instead of fixed distance FRET-based annihilation between chromophores. In addition, the method offers facile differentiation between J- and H-type aggregates, determining valuable material properties such as the exciton diffusion length, the dimensionality of diffusion and the degree of nanoscale disorder in the aggregate. The psTRAB technique therefore offers valuable opportunities to explore the nanoscale organization and excitonic coupling of chromophores in light-emitting materials with unprecedented detail.

## Methods

**Photon correlation, data analysis, and derivation of Eq. (2)**. The psTRAB is computed from raw time-stamped TCSPC data using MATLAB. The scripts developed operate similarly to conventional calculations of cross-correlations[46]. The following parameters are stored for each photon event: (i) the "macrotime" at which the photon arrived, i.e. the integer multiple of the corresponding excitation laser repetition period $T$; (ii) the "microtime", $t$, which corresponds to the time the photon was detected after the excitation pulse excited the NP; and (iii) the detection channel, i.e. the photon counter $A$ or $B$. The events are cross-correlated with respect to their macrotimes, after which the microtimes are evaluated as follows: (i) we store the shorter microtime, $t$, of each correlation event (e.g. the

microtime of channel $A$) and neglect the longer microtime, $t + \Delta t$. (ii) For selected microtime intervals, histograms of correlation events are constructed as a function of the macrotime delay between the channels. Finally, the scripts sum over multiple measurements of individual aggregates to produce an overall psTRAB result. As detailed in the Supplementary Information, we rationalize the number of correlation events, $N_c(t, t + \Delta t)$, for a given delay time $\Delta t < T - t$ between two photon events arising from the same excitation pulse, as follows:

$$N_c(t, t + \Delta t) = N_{exc} \cdot P(t) \cdot P'(t + \Delta t) \qquad (4)$$

Here, $N_{exc}$ is the total number of observed laser excitation pulses, $P(t)$ is the probability of detecting the first photon at microtime $t$ and $P'(t + \Delta t)$ is the probability of detecting the second photon at microtime $t + \Delta t < T$. In case the exciton annihilation is determined by a single exponential decay rate $k_{SSA}$, these probabilities are calculated as

$$P(t) = n_{dyes} p_0 e^{-(k_r + k_{nr} + k_{ET})t} \qquad (5)$$

$$P'(t + \Delta t) = (n_{dyes} - 1) p_0 e^{-(k_r + k_{nr} + k_{ET})t} e^{-(k_r + k_{nr})\Delta t}, \qquad (6)$$

where $n_{dyes}$ is the number of chromophores, $p_0$ summarizes the probability of the chromophore being excited by the laser pulse and the probability of detecting the emitted photon, $k_r$ and $k_{nr}$ are the radiative and non-radiative decay rates and $k_{ET} = k_{SSA}/2$ is the energy-transfer rate between two excited chromophores. Note that in general $P'(t) \neq P(t)$ since the exciton emitting the first photon at time $t$ can reside on any one of the $n_{dyes}$ chromophores, while the exciton emitting the second photon resides on one of the $(n_{dyes} - 1)$ remaining chromophores. At microtime delays $0 < \Delta t < T - t$, the number of excitons does not decay any further through energy transfer, since only a single exciton is left. The number of correlation events $N_\ell(t, t + \Delta t)$, where the second photon is detected at non-zero macrotime delays and thus arises due to a separate laser excitation event, is instead calculated from

$$N_\ell(t, t + \Delta t) = N_{exc} \cdot P''(t) \cdot P''(t + \Delta t), \qquad (7)$$

where

$$P''(t) = n_{dyes} p_0 e^{-(k_r + k_{nr})t} \qquad (8)$$

is independent of energy transfer, since only single excitons are present after each laser excitation. The ratio $N_c/N_\ell$ of central to lateral correlation events is thus directly connected to the number of chromophores in the mcNP and the time dynamics of the annihilation process as

$$\frac{N_c}{N_\ell} = \frac{n_{dyes}(n_{dyes} - 1)}{n_{dyes}^2} e^{-k_{SSA}t} = \frac{n_{dyes} - 1}{n_{dyes}} e^{-k_{SSA}t} \qquad (9)$$

The result is independent of $k_{nr}$ implying that additional quenching processes due to singlet–triplet annihilation or the interaction of singlet excitons with dark states such as charge-separated states do not impact the ratio $N_c/N_\ell$. Note that the result is independent of $\Delta t$ and it can also be calculated from the time-integrated number of correlations

$$N_c(t) = \int_0^{T-t} N_c(t, t + \Delta t) d(\Delta t), \; N_\ell(t) = \int_0^{T-t} N_\ell(t, t + \Delta t) d(\Delta t) \qquad (10)$$

which significantly reduces the noise associated with experimental event data.

Comparing the derived expression for $N_c/N_\ell$ with Eq. (1) defining the number of independent chromophores $n$, we obtain

$$n(t) = \left( 1 - \frac{n_{dyes} - 1}{n_{dyes}} \exp(-k_{SSA}t) \right)^{-1}. \qquad (11)$$

Equation (11) corresponds to Eq. (2) with $y_0 = 1$ and $A = 1 - n_{dyes}^{-1}$. A quantum-statistical description of photon correlations in an $n$-chromophore system, using a master equation approach, is given in the Supplementary Information together with Supplementary Figs. 11–13. Note that the assumption of any specific decay law for singlet–singlet annihilation such as an exponential decay according to $e^{-k_{SSA}t}$ is not strictly necessary. To that end, psTRAB $N_c/N_\ell$ can be used to directly measure the decay law associated with exciton–exciton interactions, which is connected to the mean first passage time of the random walk performed by the excitons. The technique can obviously be extended to higher-order photon correlations, using more than one beam splitter in the Hanbury Brown and Twiss setup, to determine the functional difference between two-exciton interactions and higher-order contributions.

**DNA origami microscopy.** A custom-made confocal microscope based on an Olympus IX-71 inverted microscope was used. Multichromophoric DNA-origami structures (see Supplementary Information for details on DNA–origami structures and a complete list of all primers used in Supplementary Table 4) were excited by a pulsed laser (636 nm, ~80 ps full-width half-maximum, 80 MHz, LDH-D-C-640; PicoQuant GmbH) operated at 40 MHz repetition rate. Circularly polarized light was obtained by a linear polarizer (LPVISE100-A, Thorlabs GmbH) and a quarter-wave plate (AQWP05M-600, Thorlabs GmbH). The light was focused onto the sample by an oil-immersion objective (UPLSAPO100XO, NA 1.40, Olympus Deutschland GmbH). The sample was moved by a piezo stage (P-517.3CD, Physik

Instrumente (PI) GmbH & Co. KG) controlled by a piezo controller (E-727.3CDA, Physik Instrumente (PI) GmbH & Co. KG). The emission was separated from the excitation beam by a dichroic beam splitter (zt532/640rpc, Chroma) and focused onto a 50-µm pinhole (Thorlabs GmbH). The emission light was separated from scattered excitation light by a 647 nm long-pass filter (RazorEdge LP 647, Semrock) and split into two detection channels by a non-polarizing 50:50 beam splitter (CCM1-BS013/M, Thorlabs GmbH). In each detection channel, afterglow of the avalanche photodiode was blocked by a 750 nm short-pass filter (FES0750, Thorlabs GmbH). Emission was focused onto avalanche photodiodes (SPCM-AQRH-14-TR; Excelitas Technologies GmbH & Co. KG) and signals were registered by a multichannel picosecond event timer (HydraHarp 400, PicoQuant GmbH). The setup was controlled by a commercial software package (SymPhoTime64, Picoquant GmbH).

**PPEB aggregate microscopy.** Single polymer aggregates were measured on a custom-designed confocal microscope as described elsewhere[34]. For excitation, the frequency-doubled output of a Ti:Sapphire oscillator (~100 fs, 80 MHz, 810 and 880 nm) (Chameleon, Coherent) was used, centred at 405 nm for PPEB-1 and 440 nm for PPEB-2. Femtosecond excitation was required to ensure that double excitation of the aggregates did not occur, because the excited state lifetime for the J-type coupled PPEB-2 aggregates is significantly shorter than for the DNA-origami dyes[18], preventing the use of conventional picosecond laser diodes. The laser was spatially expanded, spectrally cleaned and coupled into the microscope base (IX71, Olympus Deutschland GmbH), where it filled the backplane of a ×60 1.35 NA objective (UPLSAPO60XO, Olympus Deutschland GmbH). The sample was placed on a piezo stage (P-527.3CL, Physik Instrumente GmbH, Germany), which was scanned to generate microscope images and locate individual aggregates. The PL was detected using two single-photon detectors (PD-25-CTE, Micro Photon Devices S.r.l., Italy) connected to a multichannel picosecond event timer (HydraHarp 400, PicoQuant GmbH, Germany) allowing TCSPC and cross-correlations to be performed. The piezo stage and photon counting hardware were controlled using a customized code in LabVIEW (National Instruments).

**Exciton diffusion in PPEB aggregates.** Bulk exciton–exciton annihilation by SSA is conventionally described by a simple second-order reaction equation, $\frac{d}{dt}\rho_{exc} = -\gamma(t)\rho_{exc}^2$, where $\rho_{exc}$ is the exciton density and $\gamma(t)$ is the diffusion-controlled annihilation rate. In the context of our psTRAB method, differentiation of Eq. (11) ultimately leads to

$$\frac{d}{dt} n = -k_{SSA} \cdot n(n-1). \qquad (12)$$

for the number of independent chromophores. This function is the correct form of the second-order reaction equation in cases where the number of reactants is low, since the reaction rate of change is proportional to the number of pairs that can be chosen. The psTRAB measurements thus resolves SSA on the single-nanoparticle level in a form that can be thought of qualitatively as tracking the mutual annihilation of independent chromophores by bimolecular interaction. From Eq. (12), we derive the following linear form governing the exciton annihilation rate $\gamma = k_{SSA} V_{agg}$, where $V_{agg}$ is the aggregate volume:

$$-V_{agg} \cdot \ln\left( \frac{n-1}{n} \right) = \gamma \cdot t - V_{agg} \cdot \ln\left( \frac{n_0 - 1}{n_0} \right). \qquad (13)$$

See the Supplementary Information for details on how $V_{agg}$ is obtained by simply invoking knowledge of the mass and mass density of the polymer chain and the number of chains in the aggregate. Thus, plotting $\ln\left(\frac{n}{n-1}\right) \cdot V_{agg}$ as a function of $t$ as in Fig. 4c, e allows $\gamma$ to be determined from the gradient by straightline fitting.

## Data availability
All relevant data are available from the authors.

## Code availability
All relevant codes to analyse the data are available from the authors

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

## Acknowledgements

T.E. thanks the Deutsche Forschungsgemeinschaft (German Research Foundation) for funding through Collaborative Grant No. 319559986. F.J.H. thanks the Deutsche Forschungsgemeinschaft (DFG, German Research Foundation) for funding through the postdoctoral "booster program" of—Project-ID 314695032—S.F.B. 1277, Project B03. P.T., F.S. and T.S. thank the European Union's Horizon 2020 research and innovation programme under grant agreement No 737089 (Chipscope) and the DFG under Germany's Excellence Strategy—EXC 2089/1—390776260 for financial support. We thank Dr. Florian Selbach for TEM imaging of the samples.

## Author contributions

G.J.H., F.S., F.J.H. and J.V. devised the psTRAB methodology and developed the technique. T.S., J.V. and P.T. designed the DNA origami structures. T.S. prepared, measured and analysed the DNA origami structures and data. D.L., K.R. and S.H. designed and synthesized the conjugated polymers. G.J.H., F.S. and T.E. measured and analysed the PPEB aggregate data. S.B. developed the analytic and quantum-statistical treatment of psTRAB. G.J.H., T.S., F.S., S.B., P.T., J.M.L. and J.V. contributed to manuscript writing.

## Funding

## Competing interests

The authors declare no competing interests.
