## [Peer Review File · Nature Communications]

REVIEWER COMMENTS

Reviewer #1 (Remarks to the Author):

Hedley et al. report a novel approach to determine the motion of excitons, singlet-singlet annihilation, and the true number of fluorophores by use of time-resolved photon-antibunching measurements of fluorescent single-molecule samples. The concept is based on the statistical analysis of several hundreds of single-molecule samples (DNA-origami or photo-luminescent polymers) recorded by time-correlated single-photon counting (TCSPC). The authors were the first to use the microtime information (which is usually used to measure excited state lifetimes) for quantitatively determining the antibunching effect at different times on the picoseconds time scale. This approach allows them to determine the kinetics of photo-physical and photo-chemical processes affecting the number of emitting fluorophores on the picoseconds time scale. The authors use their approach to study the effects of exciton motion and singlet-singlet annihilation in different samples. Importantly, the authors provide a systematic investigation of the distance dependency of said processes by use of DNA origami which allow a precise control of the distance between the fluorophores and enable quantification of the kinetics of the underlying processes. Finally, they demonstrate successful application of their approach for the study of photo-physical processes in photo-luminescent polymer aggregates which provide deep insights into the molecular interactions of this important class of materials.

I can only but congratulate the authors to their impressive work which I found very exciting to read. I can see a great potential of their approach for further applications where photo-physical interactions can be studied on the microscopic scale (where most if not all photo-physical processes happen anyway). The manuscript was easy to read and will most likely attract a broad range of readers in the fields of spectroscopy and material sciences and also in biophysics. The experimental design is sound and the authors did a careful and qualified interpretation of their data.

As such I'm happy to recommend publication of the manuscript after a few minor revisions which I've listed below. I'm convinced that no further review will be required for this.

Recommended minor revisions:

- p.5; line 94 states fluorophores are represented by 'black disks' in fig. 1 while they are actually white.
- p.5; fig. 1c: the authors might consider to colour both fluorophores undergoing SSA as now it looks more like SSA is an interaction between one fluorophore in ground and the other in excited state.
- p.5; eq. 1 explains how antibunching is quantified without accounting for background which is however explained in the supplemental information (section 4). It would be advisable if the authors add a brief statement on the background contribution in the manuscript with reference to the resp. section in the SI.
- p.9; line 192 onwards: The authors state that the peak of the IRF is taken to determine zero microtime. However, the finite width of the IRF also implies an error in time which is not accounted for in the data/data analysis. It would be advisable to take the timing error into account as well.
- p.10; fig. 3: Readability of fig. 3c,d would be improved if the distances between the fluorophores would be stated in the figure in the same way as in fig. 3a,b.
- p.11/12 discusses the effect of homo-FRET for SSA with focus on the nominal distances between the labelling sites on the DNA origami. However, the nominal distance also comes with an error with the fluorophores being attached to the oligonucleotides via a linker. Assuming experiments were carried out in buffered solution this adds some variance in (i) the distance between the fluorophores and (ii) the relative orientations of their resp. transition dipole moments owing to the flexibility of the linkers. Both

effects are not discussed in the manuscript. While the orientational effects might play a minor role in the DNA origami, they might become prominent in the later experiments in polymers where I would expect the orientations to be rather fixed on the experimental time scale. A thorough discussion of these effects in the context of the experiments would be informative for the interested reader.

Reviewer #3 (Remarks to the Author):

This paper uses an unique approach of essentially time gating the antibunching signal to measure how it evolves from that characteristic of multiple emitters to that characteristic of a single emitter due to energy transfer within a multi-chromophoric structure followed by singlet-singlet annihilation (SSA) which reduces the number of emissive species. I very much like this methodology and think it can have many applications beyond those shown here and therefore is of general interest. The authors use a series of DNA origami structures in which dyes are placed at well defined distances from one another in order to show the efficacy of the method and its ability to 'count' the number of dyes that are placed on the structure. Finally, the energy transfer and SSA properties of H and J aggregates of a conjugated polymer are compared. Below I detail some areas in which I believe additional data and/or clarification would be needed and useful. Very generally speaking, the manuscript is very light on primary raw data and the bulk of my suggestions are places in which I feel this should be shown to the reader.

1. P 8-9 and SI fig S3. The question of how photobleaching will affect the measurements immediately arises in the reader's mind. The authors should say in the main text how they have made sure photo bleaching doesn't affect the results rather than just refer to S3. Analogous traces to S3 should be shown in the SI for the H and J aggregates as these could easily have different propensities for photo bleaching.
2. There should be power dependence of the fluorescence emission intensity shown as SSA processes should demonstrate a sub-linear dependence. The authors say they are in the low power regime- I didn't see a calculation of the excitation rate or, better yet, a power dependence for any of the signals. Figure 2a and p. 11 mentions multiple excitations of a single dye within the same laser pulse.
3. P. 9-10 The distinction between ensemble and single molecule experiments in probing SSA that is being made is not completely clear to me. In both cases, one needs to excite multiple entities within a structure to undergo SSA if I am not mistaken. Are the authors saying that higher powers are needed in ensemble measurements in order to see SSA in the decay dynamics due to a S/N constraint? I am not sure it is accurate to say that (top p 10) "...it is crucial to realize that SSA always occurs, even at the lowest excitation fluences, because exciton diffusion always occurs". There is always rate competition between radiative and non-radiative decay and exciton diffusion not to mention different propensities for SSA among chromophores that need to be accounted for.
4. Figure 2. Can some raw data traces of the antibunching be shown in all cases (even if in the SI). Particularly in the polymers, are the single chains showing no central peak whatsoever?
5. The description of eq. (3) is somewhat unclear to me. I understand that the rates of decay are a combination of exciton diffusion and SSA but the equation shown seems to indicate two rates of SSA judging by the subscripts. Does this mean that diffusion and SSA are captured within a single constant?
6. P. 13 In conjugated polymers (more generally) chemical or photo-chemically produced defects can play the same role they may play in the DNA structures, that is lead to quenching of excited states. Can the authors say something about this and show what is done to minimize this effect on the data?
7. P. 15 The reader may appreciate a sentence or two describing one vs two dimensional diffusion and why it may be significant in these systems.

8. P.16 The authors attribute the differences in SSA rates in H vs J aggregates to faster diffusion in H aggregates and argue why the ordered chain structure in H aggregates favors faster energy transfer. However, my understanding is that one can think of SSA itself as a homoFRET type process so should also be favored by parallel alignment of chains ect. Is there a way to distinguish these mechanisms in the author's formalism?

9. I am a little perplexed by figure S6. The caption suggests these spots are all single chains but their size varies considerably throughout the image with some objects being close to a micron in size (images could use a scale bar and a more sensible way of dividing up the axes so that each tic mark is a whole number). That is more or less the case in Fig S7 as well though there are fewer small objects. Are the authors saying that all the small individual chains (dispersed among the aggregates) in S6 coalesce into aggregates in figures S7?

In summary, I like the technique and the systems are interesting. Several aspects of the paper could use clarification and it would benefit from showing more raw data.

Reviewer 1:

Hedley et al. report a novel approach to determine the motion of excitons, singlet-singlet annihilation, and the true number of fluorophores by use of time-resolved photon-antibunching measurements of fluorescent single-molecule samples. The concept is based on the statistical analysis of several hundreds of single-molecule samples (DNA-origami or photo-luminescent polymers) recorded by time-correlated single-photon counting (TCSPC). The authors were the first to use the microtime information (which is usually used to measure excited state lifetimes) for quantitatively determining the antibunching effect at different times on the picoseconds time scale. This approach allows them to determine the kinetics of photo-physical and photo-chemical processes affecting the number of emitting fluorophores on the picoseconds time scale. The authors use their approach to study the effects of exciton motion and singlet-singlet annihilation in different samples. Importantly, the authors provide a systematic investigation of the distance dependency of said processes by use of DNA origami which allow a precise control of the distance between the fluorophores and enable quantification of the kinetics of the underlying processes. Finally, they demonstrate successful application of their approach for the study of photo-physical processes in photo-luminescent polymer aggregates which provide deep insights into the molecular interactions of this important class of materials.

I can only but congratulate the authors to their impressive work which I found very exciting to read. I can see a great potential of their approach for further applications where photo-physical interactions can be studied on the microscopic scale (where most if not all photo-physical processes happen anyway). The manuscript was easy to read and will most likely attract a broad range of readers in the fields of spectroscopy and material sciences and also in biophysics. The experimental design is sound and the authors did a careful and qualified interpretation of their data.

As such I'm happy to recommend publication of the manuscript after a few minor revisions which I've listed below. I'm convinced that no further review will be required for this. Recommended minor revisions:

- p.5; line 94 states fluorophores are represented by 'black disks' in fig. 1 while they are actually white.

R1.1: We thank the reviewer and changed the text accordingly.

- p.5; fig. 1c: the authors might consider to colour both fluorophores undergoing SSA as now it looks more like SSA is an interaction between one fluorophore in ground and the other in excited state.

R1.2: This a good suggestion and we changed fig. 1c so that the two chromophores undergoing SSA are coloured.

- p.5; eq. 1 explains how antibunching is quantified without accounting for background which is however explained in the supplemental information (section 4). It would be advisable if the authors add a brief statement on the background contribution in the manuscript with reference to the resp. section in the SI.

R1.3: We added a sentence explaining that the background is not accounted for in eq. (1) and refer to ref. 15 and the corresponding section in the SI, pointing out that the background must be taken into consideration.

- p.9; line 192 onwards: The authors state that the peak of the IRF is taken to determine zero microtime. However, the finite width of the IRF also implies an error in time which is not accounted for in the data/data analysis. It would be advisable to take the timing error into account as well.

R1.4: The timing error for a single detected photon would be given by the standard deviation, i.e. the width of the IRF. However, we accumulate the statistics over multiple detected photons in a certain time window, e.g. from 200-400 ps after excitation, and therefore the standard error of the mean must be taken as the timing error, which is negligible for the count rates obtained here. This is a standard procedure for every PL decay obtained by TCSPC, for which the timing errors on the x-axis are defined by the binning. We state now in the manuscript that the width of the binning defines the timing error.

- p.10; fig. 3: Readability of fig. 3c,d would be improved if the distances between the fluorophores would be stated in the figure in the same way as in fig. 3a,b.

R1.5: We changed fig. 3 accordingly, thank you.

- p.11/12 discusses the effect of homo-FRET for SSA with focus on the nominal distances between the labelling sites on the DNA origami. However, the nominal distance also comes with an error with the fluorophores being attached to the oligonucleotides via a linker. Assuming experiments were carried out in buffered solution this adds some variance in (i) the distance between the fluorophores and (ii) the relative orientations of their resp. transition dipole moments owing to the flexibility of the linkers. Both effects are not discussed in the manuscript. While the orientational effects might play a minor role in the DNA origami, they might become prominent in the later experiments in polymers where I would expect the orientations to be rather fixed on the experimental time scale. A thorough discussion of these effects in the context of the experiments would be informative for the interested reader.

R1.6: We added a short paragraph, which discusses the impact of transition dipole orientations regarding energy transfer in ordered vs. disordered conjugated polymer aggregates.

Reviewer 2:

In the manuscript by Hedley et al., a new method is introduced to extract the number of independent emitters as a function of time (after excitation). To do so, they construct time-resolved anti-bunching histograms. This innovative idea extends the use of TC-SPC and antibunching in order to extract rates of excited state annihilation events and see how the number of independent emitters evolves in time. The manuscript is very clear and nice model measurements are presented on well-defined origami systems with well-defined distances and number of chromophores in order to showcase the potential of the method. In the second part, a more realistic/chaotic conjugated polymer system is analyzed in order to showcase the potential of the technique in the vast realm of material and life sciences. Overall I highly suggest the publication of this important work. I have a few minor comments that could improve the clarity of the work.

Comments.

1) Perhaps a few words could be added explaining the rationale between choosing time windows of 200 ps. Does the IRF play a role here and how is the accuracy towards the end of the decay? Are there still enough photons left to determine the number of independent emitters accurately?

R2.1: The reviewer raises an important technical detail here. The width of the time windows of 200 ps is determined predominantly by the number of photons available, i.e. the photon “budget” for generating a meaningful photon antibunching histogram. Of course, the smaller the time window the higher the time resolution, but the error for determining n increases. For this reason, one must choose the time windows accordingly for each experiment and available photon budget. In our case, 200 ps was sufficient for the DNA origami model structures, but for the H- and J-type aggregates we used different time windows appropriate for the rate of change of n with time and the overall photon budget. We started with the smallest time window at early times, set to a mere 3 ps. Note that we do not claim that this is the time *resolution* of the experiment. As the reviewer correctly points out, the IRF is the actual limitation of the time resolution, but we simply plot the dynamics with a higher point density than the IRF (much as one does in conventional TCSPC) because the photon budget allows it. Note also that we have been cautious in interpreting all early (< 125 ps) psTRAB dynamics and have only made clear assignments, and the calculations of diffusion coefficients at longer times, where it is safe to do so. The time window at late times for the PPEB aggregates, where n is almost constant, was set to 80 ps for the H-type aggregate and 160 ps for the J-type aggregate. The photon budget can, in principle, be increased arbitrarily by measuring over an extended period and accumulating the photons over multiple single nanoparticles. It is therefore also possible, though not always practical, to decrease the length of the time windows arbitrarily at the expense of overall measurement time. We added a short paragraph to explain this trade-off between the choice of the length of the time windows and the error in determining n . Finally, the accuracy to determine n towards the end of the fluorescence decay transient can be estimated by the noise on the data points in Figure 4b. It turns out that, in the present case, this noise is still sufficient to extract the exciton diffusion length.

2) On page 10, the authors make a comment that the technique can be used at arbitrarily weak fluencies at the cost of extended integration times. That statement might be a bit optimistic. I would guess that the number of coincident events is not linearly dependent on the count rate.

R2.2: The reviewer is of course correct that the number of coincidence events is not linearly dependent on the count rate. In fact, it depends on the square of the count rate. However, the type of dependency does not matter if the number of coincidence events increases with increasing measurement time. But to be fair, at some point we are limited by the background photons, because the actual signal might not be distinguishable from the background anymore. The background might be very low for late microtimes because scattering photons are not detected so long after the excitation pulse, but the dark count rate of the photodetectors still plays a role. We have thus softened our statement to read: *“Our photon correlation technique is sensitive precisely and only to these rare events of double-chromophore excitation, which can be reached at very weak fluences at the cost of extended integration times. The detection of these rare events is ultimately limited by the background photons, e.g. the dark count rate of the photodetectors.”*

3) On page 12, line 270, the second decay component is ascribed to a combination of direct annihilation and hopping (with the latter being the rate determining step). Why is long range annihilation not considered? Is there evidence that the hopping rate is indeed faster than the direct long range annihilation?

R2.3: Please see response R3.5. The first rate $k_{SSA,1}$ describes the direct annihilation between neighbouring dyes, whereas $k_{SSA,2}$ describes the combined rate of a hopping process with subsequent annihilation between neighbouring dyes **and** the direct annihilation between next-nearest-neighbouring dyes. Therefore, the long-range annihilation is considered here in the rate k_{SSA}^2 . We changed the wording in the manuscript regarding eq. (3) to clarify this point.

4) Regarding both conjugated polymers, why is it that only SSA is assumed to be the main exciton annihilator? What about singlet triplet or singlet dark state annihilation, singlet defect quenching, ... The authors could show the actual decay traces of both systems to see if quenching plays a role. For the H-aggregate, this should be the clearest case since one expects the H aggregate to have a decay time much longer than the monomer. If it is similar or shorter than the monomer, quenching processes might play a role.

R2.4: The reviewer is right that the H-type aggregate should have a decay time much longer than the monomer. We previously described this expected phenomenon in the materials used in the present experiment, please see our previous work in refs. 2 and 18 as cited in manuscript. Figure 3e in ref. 2 shows the PL decays of the H-type aggregates (red) in comparison with the single chains (green).

The PL lifetime is an order of magnitude longer as compared to the single chains, indicating that exciton quenching processes, for example by defects, do not play a role on these time scales. However, even if singlet exciton quenching by triplet excitons or by other dark states such as charge-separated states occurs it will not impact the psTRAB results since these are only

additional non-radiative decay channels, which only serve to increase k_{nr} in equations (5), (6) and (7). The number of independent emitters is calculated by the ratio N_c/N_ℓ and the non-radiative rate k_{nr} cancels out (see eq. (9)). We added a short statement to this effect in the methods section.

Reviewer 3:

This paper uses a unique approach of essentially time gating the antibunching signal to measure how it evolves from that characteristic of multiple emitters to that characteristic of a single emitter due to energy transfer within a multi-chromophoric structure followed by singlet-singlet annihilation (SSA) which reduces the number of emissive species. I very much like this methodology and think it can have many applications beyond those shown here and therefore is of general interest. The authors use a series of DNA origami structures in which dyes are placed at well defined distances from one another in order to show the efficacy of the method and its ability to ‘count’ the number of dyes that are placed on the structure. Finally, the energy transfer and SSA properties of H and J aggregates of a conjugated polymer are compared. Below I detail some areas in which I believe additional data and/or clarification would be needed and useful. Very generally speaking, the manuscript is very light on primary raw data and the bulk of my suggestions are places in which I feel this should be shown to the reader.

1. P 8-9 and SI fig S3. The question of how photobleaching will affect the measurements immediately arises in the reader’s mind. The authors should say in the main text how they have made sure photo bleaching doesn’t affect the results rather than just refer to S3. Analogous traces to S3 should be shown in the SI for the H and J aggregates as these could easily have different propensities for photo bleaching.

R3.1: We thank the reviewer for raising this important point and we agree that in standard photon antibunching experiments photobleaching and blinking will have an impact on the antibunching results. However, since our measurement cycles over many single excitation laser pulses, it effectively averages the photon statistics over the course of the complete measurement. For this reason, the decay dynamics of a psTRAB curve will not be affected by photobleaching, even though the overall amplitude, i.e. the overall strength of photon antibunching, may be influenced. However, our results on the DNA origami model systems suggest that photobleaching and blinking are not an issue, because in each measurement we do indeed reach the correct number of physical dye molecules attached to the DNA origami for early times after excitation, e.g. 4.7 ± 0.2 for the five-dye structure. For the H- and J-type aggregates the PL traces are similarly stable, which lets us conclude that the amplitude of the psTRAB curves are also not affected by photobleaching and blinking. We added a paragraph discussing the influence of photobleaching and how we avoid the impact of it in our experiments. Additional PL traces of the H- and J-aggregates are now given in the SI (Figure S8).

2. There should be power dependence of the fluorescence emission intensity shown as SSA processes should demonstrate a sub-linear dependence. The authors say they are in the low power regime- I didn’t see a calculation of the excitation rate or, better yet, a power dependence for any of the signals. Figure 2a and p. 11 mentions multiple excitations of a single dye within the same laser pulse.

R3.2: We thank the reviewer for this insightful suggestion and have undertaken a brief series of power dependency measurements to explore this proposal. Working with the H-aggregate

PPEB-1, we measured at ~ 1.8 and $\sim 3.6 \mu\text{J cm}^{-2}$ fluence (determined from the power at the objective and the size of the diffraction limited spot) and find that the psTRAB decays show identical behaviour, i.e. SSA is the same in both cases. This new result has been added to the SI (Figure S9). The apparent independence on power can be rationalized by investigating the statistics of photon coincidence. Instances where we have two excitons in the aggregate such that SSA can occur with this technique are very rare. By virtue of the single-photon counting that we carry out we can identify these rare events, and we find that it is on the order of 30 instances where we record two photons per hundred thousand instances where we record one (i.e. 300 ppm of all events). Thus, we are in an excitation regime where annihilation is actually very rare, and consequently the sub-linear fluorescence power dependence that the reviewer seeks will not be detectable because there is almost always only one exciton present in the aggregate at any one time. In other words, the correction of SSA to the TCSPC PL transient amounts to 300 ppm under these excitation conditions, which is actually smaller than the shot noise ($\sqrt{N}/N = \sim 1000$ ppm) under the given measurement conditions. This seeming “discrepancy” does, however, demonstrate the power of the psTRAB technique, as we can monitor those very rare events with relative ease.

3. P. 9-10 The distinction between ensemble and single molecule experiments in probing SSA that is being made is not completely clear to me. In both cases, one needs to excite multiple entities within a structure to undergo SSA if I am not mistaken. Are the authors saying that higher powers are needed in ensemble measurements in order to see SSA in the decay dynamics due to a S/N constraint? I am not sure it is accurate to say that (top p 10) “...it is crucial to realize that SSA always occurs, even at the lowest excitation fluences, because exciton diffusion always occurs”. There is always rate competition between radiative and non-radiative decay and exciton diffusion not to mention different propensities for SSA among chromophores that need to be accounted for.

R3.3: In combination with the previous point, this is an important distinction that the reviewer makes, which we have now clarified in the manuscript (page 16). In ensemble measurements, one is going to need an appreciable fraction of excitons to undergo SSA for this to become visible in the ensemble time-resolved PL, where the signal is comprised of both excitons that have undergone SSA and those which have not; the net result in the ensemble will be a reduction of effective PL lifetime with increasing fluence. Typically, as the reviewer indicates, this condition necessitates ensemble measurements to be made at higher powers to reach a sufficiently high fraction of SSA in the exciton population such that it becomes detectable in the ensemble. With our psTRAB technique we can now operate at any power, right down to levels where SSA almost never occurs, but because we are measuring the two-photon coincidences we can still observe these very rare events background-free. Of course, we may have to measure for longer times to obtain an acceptable signal-to-noise ratio, but what psTRAB essentially does is trade acquisition fluence for acquisition time, and in doing so it gives us unique access to the low-fluence behaviour of SSA – which is actually the most relevant to application conditions. One notes as an aside that in ensemble SSA measurements, one cannot revert to ever higher fluences to obtain a better signal-to-noise ratio, since SSA simply becomes ever faster at higher exciton densities, making it ever harder to detect.

4. Figure 2. Can some raw data traces of the antibunching be shown in all cases (even if in the SI). Particularly in the polymers, are the single chains showing no central peak whatsoever?

R3.4 We are happy to do this and have added representative antibunching graphs to the SI for the polymers (Figure S10). At late times, the PPEB aggregates do indeed show almost no central peak, implying that fluorescence photons are almost perfectly “antibunched”.

5. The description of eq. (3) is somewhat unclear to me. I understand that the rates of decay are a combination of exciton diffusion and SSA at the equation shown seems to indicate two rates of SSA judging by the subscripts. Does this mean that diffusion and SSA are captured within a single constant?

R3.5: The reviewer is right according to eq. (3). The first rate, k_{SSA}^1 , describes the rate with which two neighbouring dye molecules annihilate excitons, whereas the second rate, k_{SSA}^2 , determines the combination of direct annihilation of two next-neighbouring dyes and exciton hopping with subsequent annihilation between two neighbouring dyes. According to this description, one would expect three different rates, but the data can be described sufficiently with two rates, indicating that we cannot distinguish between these two processes (annihilation between nearest-neighbouring dyes, and exciton hopping with subsequent annihilation between neighbouring dyes). We rephrased this statement in the manuscript for clarification.

6. P. 13 In conjugated polymers (more generally) chemical or photo-chemically produced defects can play the same role they may play in the DNA structures, that is lead to quenching of excited states. Can the authors say something about this and show what is done to minimize this effect on the data?

R3.6: The PL traces were carefully selected to avoid photobleaching as well as blinking due to photochemical quenchers. We have now included additional PL traces of the conjugated polymer aggregates (see also R3.1), demonstrating sufficiently stable PL over the time used to extract the photon statistics.

7. P. 15 The reader may appreciate a sentence or two describing one vs two dimensional diffusion and why it may be significant in these systems.

R3.7: We are happy to do this, and have added the following text (on page 18): “*Indeed, we would expect the one-dimensional motion along the chain in strongly coupled J-aggregates to be more pronounced than the two-dimensional motion along chains and across the π -stacks in H-aggregates, where interchromophoric coupling can be weaker.*”

8. P.16 The authors attribute the differences in SSA rates in H vs J aggregates to faster diffusion in H aggregates and argue why the ordered chain structure in H aggregates favors faster energy transfer. However, my understanding is that one can think of SSA itself as a homoFRET type process so should also be favored by parallel alignment of chains ect. Is there a way to distinguish these mechanisms in the author’s formalism?

R3.8: Building on the previous point, we are cautious in interpreting the data, and so have chosen not to make a definitive assignment. The reviewer is correct that J-aggregation should favour SSA due to chromophore alignment on-chain over the weaker H-type coupling between chains. However, these effects will likely only be most visible at the earliest times after photoexcitation, which is beyond our time-resolution. We do see a much larger gradient in the dynamics (implying faster diffusion leading to SSA) in the J-aggregates at early times (0-125 ps, Figure 4e), but we note that the gradient is still smaller than in the H-aggregate, and thus we are likely time-resolution limited in what we can see for these earliest processes. At later times (> 500 ps), the H-aggregate shows stronger diffusion and hence SSA than the J-aggregate, as

the H-type interchromophoric exciton continues to utilise the longer-range weak coupling, while for the J-type on-chain excitation, once its immediate ordered vicinity has been explored at the earliest times, the exciton is forced to diffuse through more disordered and thus significantly weaker coupled regions on longer timescales. To better capture this point we have added the following text (page 18) regarding the early timescale to acknowledge this effect and express caution with regards to it: *“We also note that at early times (0-125 ps), in the J-aggregates a significantly stronger time-dependent gradient of the psTRAB functionality is observed, consistent with fast one-dimensional exciton motion along the chain. We are cautious with regards to over-interpreting these data, however, since such exciton motion is likely to be much faster than the time resolution of our experiment.”*

9. I am a little perplexed by figure S6. The caption suggests these spots are all single chains but their size varies considerably throughout the image with some objects being close to a micron in size (images could use a scale bar and a more sensible way of dividing up the axes so that each tic mark is a whole number). That is more or less the case in Fig S7 as well though there are fewer small objects. Are the authors saying that all the small individual chains (dispersed among the aggregates) in S6 coalesce into aggregates in figures S7? In summary, I like the technique and the systems are interesting. Several aspects of the paper could use clarification and it would benefit from showing more raw data.

R3.9: We note the reviewer’s comment and can offer assurance that this is simply an effect of intensity saturation of the PL image displayed, which really is an artefact which plagues every single single-molecule fluorescence image – super-resolution or not. To make the number of single chains as clear as possible in the image, we had reduced the dynamic range of the colour scale to make the molecules easily identifiable. The reviewer is correct, however, that this leads to the misinterpretation that single chains appear to be almost of micron in size, which, of course, is not the case. We have replotted the data with a more appropriate dynamic range of the colour scale to better illustrate the presence of single chains.

REVIEWERS' COMMENTS

Reviewer #1 (Remarks to the Author):

All my questions have been answered. Publish as is.

Reviewer #2 (Remarks to the Author):

The authors have addressed all the comments. I recommend publication of the revised manuscript.

Reviewer #3 (Remarks to the Author):

The authors have adequately addressed the majority of my concerns and I believe this manuscript is ready for publication.

We thank the reviewers for their detailed and insightful feedback. We are particularly delighted to see that the reviewers share our enthusiasm about our “...*impressive work*...” and also believe that this work has “...*great potential... for further applications*” and “...*suggest publication of this important work*”. We would like to thank all referees for their careful review and agree with reviewer 3 that the manuscript will benefit from the presentation of additional raw data and some further clarifications. We therefore added additional datasets in the SI, showing more raw data and providing further explanations regarding the impact of bleaching and excitation intensity. We address all points that the reviewers have raised explicitly in the following section. A marked-up version is attached with each comment action labelled individually.

The responses are labelled by reviewer and comment number, e.g. R1.1 for reviewer 1, comment 1.

Reviewer 1:

Hedley et al. report a novel approach to determine the motion of excitons, singlet-singlet annihilation, and the true number of fluorophores by use of time-resolved photon-antibunching measurements of fluorescent single-molecule samples. The concept is based on the statistical analysis of several hundreds of single-molecule samples (DNA-origami or photo-luminescent polymers) recorded by time-correlated single-photon counting (TCSPC). The authors were the first to use the microtime information (which is usually used to measure excited state lifetimes) for quantitatively determining the antibunching effect at different times on the picoseconds time scale. This approach allows them to determine the kinetics of photo-physical and photo-chemical processes affecting the number of emitting fluorophores on the picoseconds time scale. The authors use their approach to study the effects of exciton motion and singlet-singlet annihilation in different samples. Importantly, the authors provide a systematic investigation of the distance dependency of said processes by use of DNA origami which allow a precise control of the distance between the fluorophores and enable quantification of the kinetics of the underlying processes. Finally, they demonstrate successful application of their approach for the study of photo-physical processes in photo-luminescent polymer aggregates which provide deep insights into the molecular interactions of this important class of materials.

I can only but congratulate the authors to their impressive work which I found very exciting to read. I can see a great potential of their approach for further applications where photo-physical interactions can be studied on the microscopic scale (where most if not all photo-physical processes happen anyway). The manuscript was easy to read and will most likely attract a broad range of readers in the fields of spectroscopy and material sciences and also in biophysics. The experimental design is sound and the authors did a careful and qualified interpretation of their data.

As such I'm happy to recommend publication of the manuscript after a few minor revisions which I've listed below. I'm convinced that no further review will be required for this. Recommended minor revisions:

- p.5; line 94 states fluorophores are represented by 'black disks' in fig. 1 while they are actually white.

R1.1: We thank the reviewer and changed the text accordingly.

- p.5; fig. 1c: the authors might consider to colour both fluorophores undergoing SSA as now it looks more like SSA is an interaction between one fluorophore in ground and the other in excited state.

R1.2: This a good suggestion and we changed fig. 1c so that the two chromophores undergoing SSA are coloured.

- p.5; eq. 1 explains how antibunching is quantified without accounting for background which is however explained in the supplemental information (section 4). It would be advisable if the authors add a brief statement on the background contribution in the manuscript with reference to the resp. section in the SI.

R1.3: We added a sentence explaining that the background is not accounted for in eq. (1) and refer to ref. 15 and the corresponding section in the SI, pointing out that the background must be taken into consideration.

- p.9; line 192 onwards: The authors state that the peak of the IRF is taken to determine zero microtime. However, the finite width of the IRF also implies an error in time which is not accounted for in the data/data analysis. It would be advisable to take the timing error into account as well.

R1.4: The timing error for a single detected photon would be given by the standard deviation, i.e. the width of the IRF. However, we accumulate the statistics over multiple detected photons in a certain time window, e.g. from 200-400 ps after excitation, and therefore the standard error of the mean must be taken as the timing error, which is negligible for the count rates obtained here. This is a standard procedure for every PL decay obtained by TCSPC, for which the timing errors on the x-axis are defined by the binning. We state now in the manuscript that the width of the binning defines the timing error.

- p.10; fig. 3: Readability of fig. 3c,d would be improved if the distances between the fluorophores would be stated in the figure in the same way as in fig. 3a,b.

R1.5: We changed fig. 3 accordingly, thank you.

- p.11/12 discusses the effect of homo-FRET for SSA with focus on the nominal distances between the labelling sites on the DNA origami. However, the nominal distance also comes with an error with the fluorophores being attached to the oligonucleotides via a linker. Assuming experiments were carried out in buffered solution this adds some variance in (i) the distance between the fluorophores and (ii) the relative orientations of their resp. transition dipole moments owing to the flexibility of the linkers. Both effects are not discussed in the manuscript. While the orientational effects might play a minor role in the DNA origami, they might become prominent in the later experiments in polymers where I would expect the orientations to be rather fixed on the experimental time scale. A thorough discussion of these effects in the context of the experiments would be informative for the interested reader.

R1.6: We added a short paragraph, which discusses the impact of transition dipole orientations regarding energy transfer in ordered vs. disordered conjugated polymer aggregates.

Reviewer 2:

In the manuscript by Hedley et al., a new method is introduced to extract the number of independent emitters as a function of time (after excitation). To do so, they construct time-resolved anti-bunching histograms. This innovative idea extends the use of TC-SPC and antibunching in order to extract rates of excited state annihilation events and see how the number of independent emitters evolves in time. The manuscript is very clear and nice model measurements are presented on well-defined origami systems with well-defined distances and number of chromophores in order to showcase the potential of the method. In the second part, a more realistic/chaotic conjugated polymer system is analyzed in order to showcase the potential of the technique in the vast realm of material and life sciences. Overall I highly suggest the publication of this important work. I have a few minor comments that could improve the clarity of the work.

Comments.

1) Perhaps a few words could be added explaining the rationale between choosing time windows of 200 ps. Does the IRF play a role here and how is the accuracy towards the end of the decay? Are there still enough photons left to determine the number of independent emitters accurately?

R2.1: The reviewer raises an important technical detail here. The width of the time windows of 200 ps is determined predominantly by the number of photons available, i.e. the photon “budget” for generating a meaningful photon antibunching histogram. Of course, the smaller the time window the higher the time resolution, but the error for determining n increases. For this reason, one must choose the time windows accordingly for each experiment and available photon budget. In our case, 200 ps was sufficient for the DNA origami model structures, but for the H- and J-type aggregates we used different time windows appropriate for the rate of change of n with time and the overall photon budget. We started with the smallest time window at early times, set to a mere 3 ps. Note that we do not claim that this is the time *resolution* of the experiment. As the reviewer correctly points out, the IRF is the actual limitation of the time resolution, but we simply plot the dynamics with a higher point density than the IRF (much as one does in conventional TCSPC) because the photon budget allows it. Note also that we have been cautious in interpreting all early (< 125 ps) psTRAB dynamics and have only made clear assignments, and the calculations of diffusion coefficients at longer times, where it is safe to do so. The time window at late times for the PPEB aggregates, where n is almost constant, was set to 80 ps for the H-type aggregate and 160 ps for the J-type aggregate. The photon budget can, in principle, be increased arbitrarily by measuring over an extended period and accumulating the photons over multiple single nanoparticles. It is therefore also possible, though not always practical, to decrease the length of the time windows arbitrarily at the expense of overall measurement time. We added a short paragraph to explain this trade-off between the choice of the length of the time windows and the error in determining n . Finally, the accuracy to determine n towards the end of the fluorescence decay transient can be estimated by the noise on the data points in Figure 4b. It turns out that, in the present case, this noise is still sufficient to extract the exciton diffusion length.

2) On page 10, the authors make a comment that the technique can be used at arbitrarily weak fluencies at the cost of extended integration times. That statement might be a bit optimistic. I would guess that the number of coincident events is not linearly dependent on the count rate.

R2.2: The reviewer is of course correct that the number of coincidence events is not linearly dependent on the count rate. In fact, it depends on the square of the count rate. However, the type of dependency does not matter if the number of coincidence events increases with increasing measurement time. But to be fair, at some point we are limited by the background photons, because the actual signal might not be distinguishable from the background anymore. The background might be very low for late microtimes because scattering photons are not detected so long after the excitation pulse, but the dark count rate of the photodetectors still plays a role. We have thus softened our statement to read: *“Our photon correlation technique is sensitive precisely and only to these rare events of double-chromophore excitation, which can be reached at very weak fluences at the cost of extended integration times. The detection of these rare events is ultimately limited by the background photons, e.g. the dark count rate of the photodetectors.”*

3) On page 12, line 270, the second decay component is ascribed to a combination of direct annihilation and hopping (with the latter being the rate determining step). Why is long range annihilation not considered? Is there evidence that the hopping rate is indeed faster than the direct long range annihilation?

R2.3: Please see response R3.5. The first rate $k_{SSA,1}$ describes the direct annihilation between neighbouring dyes, whereas $k_{SSA,2}$ describes the combined rate of a hopping process with subsequent annihilation between neighbouring dyes **and** the direct annihilation between next-nearest-neighbouring dyes. Therefore, the long-range annihilation is considered here in the rate k_{SSA}^2 . We changed the wording in the manuscript regarding eq. (3) to clarify this point.

4) Regarding both conjugated polymers, why is it that only SSA is assumed to be the main exciton annihilator? What about singlet triplet or singlet dark state annihilation, singlet defect quenching, ... The authors could show the actual decay traces of both systems to see if quenching plays a role. For the H-aggregate, this should be the clearest case since one expects the H aggregate to have a decay time much longer than the monomer. If it is similar or shorter than the monomer, quenching processes might play a role.

R2.4: The reviewer is right that the H-type aggregate should have a decay time much longer than the monomer. We previously described this expected phenomenon in the materials used in the present experiment, please see our previous work in refs. 2 and 18 as cited in manuscript. Figure 3e in ref. 2 shows the PL decays of the H-type aggregates (red) in comparison with the single chains (green).

The PL lifetime is an order of magnitude longer as compared to the single chains, indicating that exciton quenching processes, for example by defects, do not play a role on these time scales. However, even if singlet exciton quenching by triplet excitons or by other dark states such as charge-separated states occurs it will not impact the psTRAB results since these are only

additional non-radiative decay channels, which only serve to increase k_{nr} in equations (5), (6) and (7). The number of independent emitters is calculated by the ratio N_c/N_ℓ and the non-radiative rate k_{nr} cancels out (see eq. (9)). We added a short statement to this effect in the methods section.

Reviewer 3:

This paper uses a unique approach of essentially time gating the antibunching signal to measure how it evolves from that characteristic of multiple emitters to that characteristic of a single emitter due to energy transfer within a multi-chromophoric structure followed by singlet-singlet annihilation (SSA) which reduces the number of emissive species. I very much like this methodology and think it can have many applications beyond those shown here and therefore is of general interest. The authors use a series of DNA origami structures in which dyes are placed at well defined distances from one another in order to show the efficacy of the method and its ability to ‘count’ the number of dyes that are placed on the structure. Finally, the energy transfer and SSA properties of H and J aggregates of a conjugated polymer are compared. Below I detail some areas in which I believe additional data and/or clarification would be needed and useful. Very generally speaking, the manuscript is very light on primary raw data and the bulk of my suggestions are places in which I feel this should be shown to the reader.

1. P 8-9 and SI fig S3. The question of how photobleaching will affect the measurements immediately arises in the reader’s mind. The authors should say in the main text how they have made sure photo bleaching doesn’t affect the results rather than just refer to S3. Analogous traces to S3 should be shown in the SI for the H and J aggregates as these could easily have different propensities for photo bleaching.

R3.1: We thank the reviewer for raising this important point and we agree that in standard photon antibunching experiments photobleaching and blinking will have an impact on the antibunching results. However, since our measurement cycles over many single excitation laser pulses, it effectively averages the photon statistics over the course of the complete measurement. For this reason, the decay dynamics of a psTRAB curve will not be affected by photobleaching, even though the overall amplitude, i.e. the overall strength of photon antibunching, may be influenced. However, our results on the DNA origami model systems suggest that photobleaching and blinking are not an issue, because in each measurement we do indeed reach the correct number of physical dye molecules attached to the DNA origami for early times after excitation, e.g. 4.7 ± 0.2 for the five-dye structure. For the H- and J-type aggregates the PL traces are similarly stable, which lets us conclude that the amplitude of the psTRAB curves are also not affected by photobleaching and blinking. We added a paragraph discussing the influence of photobleaching and how we avoid the impact of it in our experiments. Additional PL traces of the H- and J-aggregates are now given in the SI (Figure S8).

2. There should be power dependence of the fluorescence emission intensity shown as SSA processes should demonstrate a sub-linear dependence. The authors say they are in the low power regime- I didn’t see a calculation of the excitation rate or, better yet, a power dependence for any of the signals. Figure 2a and p. 11 mentions multiple excitations of a single dye within the same laser pulse.

R3.2: We thank the reviewer for this insightful suggestion and have undertaken a brief series of power dependency measurements to explore this proposal. Working with the H-aggregate

PPEB-1, we measured at ~ 1.8 and $\sim 3.6 \mu\text{J cm}^{-2}$ fluence (determined from the power at the objective and the size of the diffraction limited spot) and find that the psTRAB decays show identical behaviour, i.e. SSA is the same in both cases. This new result has been added to the SI (Figure S9). The apparent independence on power can be rationalized by investigating the statistics of photon coincidence. Instances where we have two excitons in the aggregate such that SSA can occur with this technique are very rare. By virtue of the single-photon counting that we carry out we can identify these rare events, and we find that it is on the order of 30 instances where we record two photons per hundred thousand instances where we record one (i.e. 300 ppm of all events). Thus, we are in an excitation regime where annihilation is actually very rare, and consequently the sub-linear fluorescence power dependence that the reviewer seeks will not be detectable because there is almost always only one exciton present in the aggregate at any one time. In other words, the correction of SSA to the TCSPC PL transient amounts to 300 ppm under these excitation conditions, which is actually smaller than the shot noise ($\sqrt{N}/N = \sim 1000$ ppm) under the given measurement conditions. This seeming “discrepancy” does, however, demonstrate the power of the psTRAB technique, as we can monitor those very rare events with relative ease.

3. P. 9-10 The distinction between ensemble and single molecule experiments in probing SSA that is being made is not completely clear to me. In both cases, one needs to excite multiple entities within a structure to undergo SSA if I am not mistaken. Are the authors saying that higher powers are needed in ensemble measurements in order to see SSA in the decay dynamics due to a S/N constraint? I am not sure it is accurate to say that (top p 10) “...it is crucial to realize that SSA always occurs, even at the lowest excitation fluences, because exciton diffusion always occurs”. There is always rate competition between radiative and non-radiative decay and exciton diffusion not to mention different propensities for SSA among chromophores that need to be accounted for.

R3.3: In combination with the previous point, this is an important distinction that the reviewer makes, which we have now clarified in the manuscript (page 16). In ensemble measurements, one is going to need an appreciable fraction of excitons to undergo SSA for this to become visible in the ensemble time-resolved PL, where the signal is comprised of both excitons that have undergone SSA and those which have not; the net result in the ensemble will be a reduction of effective PL lifetime with increasing fluence. Typically, as the reviewer indicates, this condition necessitates ensemble measurements to be made at higher powers to reach a sufficiently high fraction of SSA in the exciton population such that it becomes detectable in the ensemble. With our psTRAB technique we can now operate at any power, right down to levels where SSA almost never occurs, but because we are measuring the two-photon coincidences we can still observe these very rare events background-free. Of course, we may have to measure for longer times to obtain an acceptable signal-to-noise ratio, but what psTRAB essentially does is trade acquisition fluence for acquisition time, and in doing so it gives us unique access to the low-fluence behaviour of SSA – which is actually the most relevant to application conditions. One notes as an aside that in ensemble SSA measurements, one cannot revert to ever higher fluences to obtain a better signal-to-noise ratio, since SSA simply becomes ever faster at higher exciton densities, making it ever harder to detect.

4. Figure 2. Can some raw data traces of the antibunching be shown in all cases (even if in the SI). Particularly in the polymers, are the single chains showing no central peak whatsoever?

R3.4 We are happy to do this and have added representative antibunching graphs to the SI for the polymers (Figure S10). At late times, the PPEB aggregates do indeed show almost no central peak, implying that fluorescence photons are almost perfectly “antibunched”.

5. The description of eq. (3) is somewhat unclear to me. I understand that the rates of decay are a combination of exciton diffusion and SSA at the equation shown seems to indicate two rates of SSA judging by the subscripts. Does this mean that diffusion and SSA are captured within a single constant?

R3.5: The reviewer is right according to eq. (3). The first rate, k_{SSA}^1 , describes the rate with which two neighbouring dye molecules annihilate excitons, whereas the second rate, k_{SSA}^2 , determines the combination of direct annihilation of two next-neighbouring dyes and exciton hopping with subsequent annihilation between two neighbouring dyes. According to this description, one would expect three different rates, but the data can be described sufficiently with two rates, indicating that we cannot distinguish between these two processes (annihilation between nearest-neighbouring dyes, and exciton hopping with subsequent annihilation between neighbouring dyes). We rephrased this statement in the manuscript for clarification.

6. P. 13 In conjugated polymers (more generally) chemical or photo-chemically produced defects can play the same role they may play in the DNA structures, that is lead to quenching of excited states. Can the authors say something about this and show what is done to minimize this effect on the data?

R3.6: The PL traces were carefully selected to avoid photobleaching as well as blinking due to photochemical quenchers. We have now included additional PL traces of the conjugated polymer aggregates (see also R3.1), demonstrating sufficiently stable PL over the time used to extract the photon statistics.

7. P. 15 The reader may appreciate a sentence or two describing one vs two dimensional diffusion and why it may be significant in these systems.

R3.7: We are happy to do this, and have added the following text (on page 18): “*Indeed, we would expect the one-dimensional motion along the chain in strongly coupled J-aggregates to be more pronounced than the two-dimensional motion along chains and across the π -stacks in H-aggregates, where interchromophoric coupling can be weaker.*”

8. P.16 The authors attribute the differences in SSA rates in H vs J aggregates to faster diffusion in H aggregates and argue why the ordered chain structure in H aggregates favors faster energy transfer. However, my understanding is that one can think of SSA itself as a homoFRET type process so should also be favored by parallel alignment of chains ect. Is there a way to distinguish these mechanisms in the author’s formalism?

R3.8: Building on the previous point, we are cautious in interpreting the data, and so have chosen not to make a definitive assignment. The reviewer is correct that J-aggregation should favour SSA due to chromophore alignment on-chain over the weaker H-type coupling between chains. However, these effects will likely only be most visible at the earliest times after photoexcitation, which is beyond our time-resolution. We do see a much larger gradient in the dynamics (implying faster diffusion leading to SSA) in the J-aggregates at early times (0-125 ps, Figure 4e), but we note that the gradient is still smaller than in the H-aggregate, and thus we are likely time-resolution limited in what we can see for these earliest processes. At later times (> 500 ps), the H-aggregate shows stronger diffusion and hence SSA than the J-aggregate, as

the H-type interchromophoric exciton continues to utilise the longer-range weak coupling, while for the J-type on-chain excitation, once its immediate ordered vicinity has been explored at the earliest times, the exciton is forced to diffuse through more disordered and thus significantly weaker coupled regions on longer timescales. To better capture this point we have added the following text (page 18) regarding the early timescale to acknowledge this effect and express caution with regards to it: *“We also note that at early times (0-125 ps), in the J-aggregates a significantly stronger time-dependent gradient of the psTRAB functionality is observed, consistent with fast one-dimensional exciton motion along the chain. We are cautious with regards to over-interpreting these data, however, since such exciton motion is likely to be much faster than the time resolution of our experiment.”*

9. I am a little perplexed by figure S6. The caption suggests these spots are all single chains but their size varies considerably throughout the image with some objects being close to a micron in size (images could use a scale bar and a more sensible way of dividing up the axes so that each tic mark is a whole number). That is more or less the case in Fig S7 as well though there are fewer small objects. Are the authors saying that all the small individual chains (dispersed among the aggregates) in S6 coalesce into aggregates in figures S7? In summary, I like the technique and the systems are interesting. Several aspects of the paper could use clarification and it would benefit from showing more raw data.

R3.9: We note the reviewer’s comment and can offer assurance that this is simply an effect of intensity saturation of the PL image displayed, which really is an artefact which plagues every single single-molecule fluorescence image – super-resolution or not. To make the number of single chains as clear as possible in the image, we had reduced the dynamic range of the colour scale to make the molecules easily identifiable. The reviewer is correct, however, that this leads to the misinterpretation that single chains appear to be almost of micron in size, which, of course, is not the case. We have replotted the data with a more appropriate dynamic range of the colour scale to better illustrate the presence of single chains.

Reviewer 1:

All my questions have been answered. Publish as is.

Reviewer 2:

The authors have addressed all the comments. I recommend publication of the revised manuscript.

Reviewer 3:

The authors have adequately addressed the majority of my concerns and I believe this manuscript is ready for publication.